# Auditing AI Models for Verified Deployment under Semantic Specifications

## Abstract

Auditing trained deep learning (DL) models prior to deployment is vital for preventing unintended consequences. One of the biggest challenges in auditing is the lack of human-interpretable specifications for the DL models that are directly useful to the auditor. We address this challenge through a sequence of semantically-aligned unit tests, where each unit test verifies whether a predefined specification (e.g., accuracy over 95%) is satisfied with respect to controlled and semantically aligned variations in the input space (e.g., in face recognition, the angle relative to the camera). We enable such unit tests through variations in a semantically-interpretable latent space of a generative model. Further, we conduct certified training for the DL model through a shared latent space representation with the generative model. With evaluations on four different datasets, covering images of chest X-rays, human faces, ImageNet classes, and towers, we show how `AuditAI` allows us to obtain controlled variations for certified training. Thus, our framework, `AuditAI`, bridges the gap between semantically-aligned formal verification and scalability. https://sites.google.com/view/audit-ai

## 1 Introduction

Deep learning (DL) models are now ubiquitously deployed in a number of real-world applications, many of which are safety critical such as autonomous driving and healthcare (Kendall et al., 2019; Miotto et al., 2018; Senior et al., 2020). As these models are prone to failure, especially under domain shifts, it is important to know when and how they are likely to fail before their deployment, a process we refer to as auditing. Inspired by the failure-mode and effects analysis (FMEA) for control systems and software systems (Teng & Ho, 1996), we propose to *audit* DL models through a sequence of semantically-aligned *unit tests*, where each unit test verifies whether a pre-defined specification (e.g., accuracy over 95%) is satisfied with respect to controlled and semantically meaningful variations in the input space (e.g., the angle relative to the camera for a face image). Being semantically-aligned is critical for these unit tests to be useful for the auditor of the system to plan the model's deployment.

The main challenge for auditing DL models through semantically-aligned unit tests is that the current large-scale DL models mostly lack an interpretable structure. This makes it difficult to quantify how the output varies given controlled semantically-aligned input variations. While there are works that aim to bring interpretable formal verification to DL models (Henriksen & Lomuscio, 2020; Liu et al., 2019), the scale is still far from the millions if not billions of parameters used in contemporary models (He et al., 2016; Iandola et al., 2014; Brown et al., 2020).

On the other hand, auditing has taken the form of verified adversarial robustness for DL models (Samangouei et al., 2018; Xiao et al., 2018a; Cohen et al., 2019). However, this has mostly focused on adversarial perturbations in the pixel space, for example, through Interval Bound Propagation (IBP), where the output is guaranteed to be invariant to input pixel perturbations with respect to $L_p$ norm (Gowal et al., 2018; Zhang et al., 2019b). While these approaches are much more scalable to modern DL architectures, the pixel space variations are not semantically-aligned, meaning they do not directly relate to semantic changes in the image, unlike in formal verification. Consider a unit test that verifies against the angle relative to the camera for

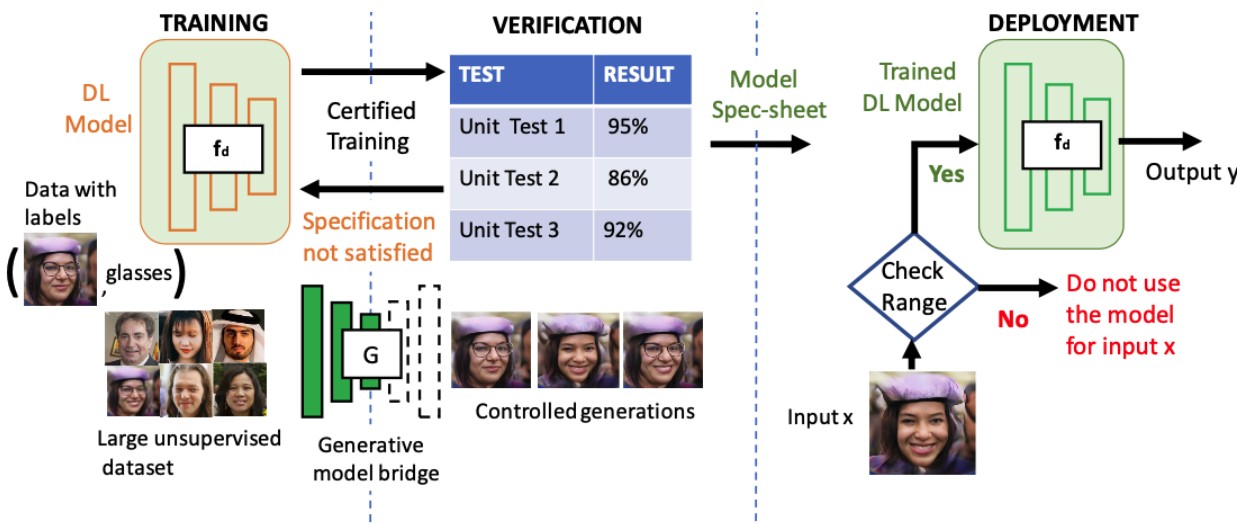

Figure 1: Basic outline of `AuditAI`. The training phase involves the design and training of the algorithms for a particular use case. The auditor lists a set of specifications that the trained model must satisfy. The auditor receives the trained model from the designer and with the set of specifications has to determine whether the model satisfies the specifications. If they are not satisfied, then the model is sent back to the designer for further updates. After that, the model is deployed, with a model spec-sheet detailing the verified range of operation. At deployment, latent embeddings of the input inform whether the input is within model-specifications. Based on the spec-sheet, it can be decided in which settings to use the model.

a face image. A small variation in the angle (e.g., facing directly at the camera versus 5° to the left) can induce a large variation in the pixel space. Current certified training methods are far from being able to provide guarantees with respect to such large variations in the pixel space with reasonable accuracy.

**Our Approach.** In order to overcome the above limitations, we develop a framework for auditing, `AuditAI`. We consider a typical machine learning production pipeline (Fig. 1) with three stages, the design and training of the model, its verification, and finally deployment. The verification is crucial in determining whether the model satisfies the necessary specifications before deployment.

We address the gap between scalability and interpretability by proposing to verify specifications for variations directly in a semantically-aligned latent space of a generative model. For example in Fig. 1, unit test 1 verifies whether a given face classification model maintains over 95% accuracy when the face angle is within $d°$, while unit test 2 checks under what lighting condition the model has over 86% accuracy. Once the verification is done, the auditor can then use the verified specification to determine whether to use the trained Deep Learning model during deployment.

For semantically-aligned latent variations, we create a bridge between the generative model and the DL model such that they share the same latent space. We incorporate a variant of IBP (Zhang et al., 2019b) for verification with respect to perturbations in the latent space. Further this leads to a tighter and a much more practical bound in the output space compared to pixel-based certified training. We also show that `AuditAI` can verify whether a unit test is satisfied by generating a proof for verification based on bound propagation. Fig. 1 gives an overview of our auditing framework and Fig. 2 elaborates on the generative model bridge.

**Summary of Contributions:**

1. We develop a framework, `AuditAI` for auditing deep learning models by creating a bridge with a generative model such that they share the same semantically-aligned latent space.
2. We propose unit tests as a semantically-aligned way to quantify specifications that can be audited.
3. We show how IBP can be applied to latent space variations, to provide certifications of semantically-aligned specifications.

We show that `AuditAI` is applicable to training, verification, and deployment across diverse datasets: ImageNet (Deng et al., 2009), Chest X-Rays (Irvin et al., 2019; Wang et al., 2017), LSUN (Yu et al., 2015), and Flicker Faces HQ (FFHQ) (Karras et al., 2019). For ImageNet, we show that `AuditAI` can train verifiably robust models which can tolerate 20% larger variations compared to pixel-based certified-training counterparts for the same overall verified error of 88% (Table 2). The variations are measured as $L_2$ distances in the pixel-space. The respective % increase in pixel-space variations that can be certified for Chest X-Rays, LSUN, and FFHQ are 22%, 19%, 24%. We conclude with a human-study of the quality of the generative model for different ranges of latent-space variations revealing that pixel-space variations upto 62% of the nominal values result in realistic generated images indistinguishable from real images by humans. Finally we describe limitations of the proposed framework that address challenges for creating a general AI safety setup for auditing arbitrary models with rich semantic variations in the scene.

## 2  `AuditAI`: A Deep Learning Audit Framework

In this section, we describe the details of our framework, `AuditAI` outlined in Fig. 1 for verifying and testing deep models before deployment. We propose to use *unit tests* to verify variations of interest that are semantically-aligned. The advantage of `AuditAI` is that the verified input ranges are given by several semantically aligned specifications for the end-user to follow during deployment. In the following sub-sections, we formally define unit tests, outline the verification approach, and describe the specifics of `AuditAI` with a GAN-bridge shown in Fig. 2.

### 2.1  Unit Test Definition

Consider a machine learning model $f : \mathcal{X} \to \mathcal{Y}$ that predicts outputs $y \in \mathcal{Y}$ from inputs $x \in \mathcal{X}$ in dataset $\mathcal{D}$. Each of our unit test can be formulated as providing guarantees such that:

$$F(x,y) \leq 0 \quad \forall x \ \ s.t. \ \ , e_i(x) \in \mathcal{S}_{i,in}, y = f(x) \tag{1}$$

Here, $i$ subscripts an unit test, encoder $e_i$ extracts the variation of interest from the input $x$, and $\mathcal{S}_{i,in}$ denotes the set of range of variation that this unit test verifies. If the condition is satisfied, then our unit test specifies that the output of the model $f(x)$ would satisfy the constraint given by $F(\cdot, \cdot) \leq 0$.

For example, $x$ could be an image of a human face, and $f$ could be a classifier for whether the person is wearing eyeglasses ($f \leq 0$ means wearing eyeglasses and $f > 0$ means not wearing eyeglasses). Then $e_i(\cdot)$ could be extracting the angle of the face from the image, and $\mathcal{S}_{i,in}$ could be the set $\{e_i(x)| \ \forall x \ |e_i(x)| < 30°\}$ constraining the rotation to be smaller than 30°. And $F(x,y) = -f(x)f(x_{0°}) \leq 0$ says that our classifier output would not be changed from the corresponding output of $x_{0°}$, the face image of $x$ without any rotation. In this case, when the end-user is going to deploy $f$ on a face image, they can first apply $e_i(\cdot)$ to see if the face angle lies within $\mathcal{S}_{i,in}$ to decide whether to use the model $f$.

### 2.2  Verification Outline

Given a specification $F(x,y) \leq 0$, we need to next answer the following questions:

1. How do we obtain the corresponding components $(e_i, S_{i,in})$ in Eq. (1)?
2. What can we do to guarantee that $F(x,y) \leq 0$ is indeed satisfied in this case?

For the sake of illustration, we first consider a scenario with a less realistic assumption. We will then discuss how we can relax this assumption. In this scenario, we assume that we are already given the encoder $e_i$ and a corresponding generator $g_i$ that inverts $e_i$. Continuing the previous example, $e_i$ can extract the face angle of a given face image. On the other hand, $g_i(x_{0°}, d°)$ would be able to synthesize arbitrary face image $x_{d°}$ that is the same as $x_{0°}$ except being rotated by $d°$.

Given the generator $g_i$ and the encoder $e_i$, we propose to obtain $\mathcal{S}_{i,in}$ building on interval bound propagation (IBP) (Gowal et al., 2018). We include a treatment of IBP preliminaries in Appendix A.1. Here, $\mathcal{S}_{i,in}$ is the largest set of variations such that the specification $F(x,y) \leq 0$ is satisfied. Given a set of variations $\mathcal{S}_{i,in}$,

IBP-based methods can be used to obtain a bound on the output of the network $\mathcal{S}_{i,out} = \{y : l_y \leq y \leq u_y\}$ and it can be checked whether the specification $F(x, y) \leq 0$ is satisfied for all values in $\mathcal{S}_{i,out}$. We can start with a initial set $\mathcal{S}_{i,in}^0$ and apply $g_i$ to this set. Then, we can first find out what would be the corresponding convex bound of variations of $\mathcal{S}_{i,in}^0$ in the image space $\mathcal{X}$. We can subsequently propagate this bound in $\mathcal{X}$ through $f$ to $\mathcal{Y}$ to get $\mathcal{S}_{i,out}^0$ and check whether $F(x, y) \leq 0$ is satisfied. By iteratively searching for the largest set $\mathcal{S}_{i,in}^j$ such that $F(x, y) \leq 0$ is still satisfied by $\mathcal{S}_{i,out}^j$, we can obtain $\mathcal{S}_{i,in}$ given $F$.

## 2.3 Certified Training through Latent Representation

In the previous section, we are able to find the set $\mathcal{S}_{i,in}$ such that Eq. (1) is satisfied given specification $F$, encoder $e_i$, generator $g_i$. However, there are several challenges that limit the practical application of this scenario. First and foremost, while significant progress has been made in generative models, especially controlled generation (Brock et al., 2018; Karras et al., 2017), the assumption of having $g_i$ apriori is still unrealistic. In addition, since the propagated bound is convex, one can imagine that the bound in $\mathcal{X}$ (high dimensional image-space) would be much larger than needed. This leads to a much narrower estimate of $\mathcal{S}_{i,in}$. While it could be true

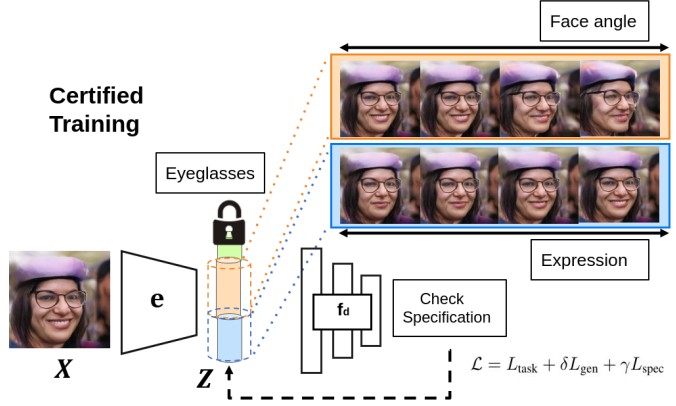

Figure 2: Instead of variations at the level of input (e.g. pixels), consider variations in the latent space. Given a specification, we use a variant of interval-bound propagation (IBP) to verify if the specification is satisfied for $\epsilon$-ball latent space variations for particular latent dimensions.

that Eq. (1) is satisfied, we still need a non-trivial size of $\mathcal{S}_{i,in}$ for the framework to be useful. For example, if $\mathcal{S}_{i,in}$ constrains face angle rotations to be within $1°$ then it may not be practical for the end-user.

For auditing ML models thoroughly, we require clear separation between the development and testing phases (i.e. black-box verification). However, it might also be desirable in practice to train the model in such a way that the unit tests are likely to be satisfied in the first place. In this case, we do have a tighter connection between the designer and the verifier. We show that by relaxing the black-box constraint, we can simultaneously address the previously listed challenges in a white-box setting.

Now that we have access to the internal working and design of the ML model $f$, we propose to bypass the image generation step in our verification process through the use of latent disentangled representations. More specifically, consider a ML model mapping $f : \mathcal{X} \rightarrow \mathcal{Z} \rightarrow \mathcal{Y}$ (denote by $f_d : \mathcal{Z} \rightarrow \mathcal{Y}$ the downstream task) and a generative model mapping $g : \mathcal{X} \rightarrow \mathcal{Z} \rightarrow \hat{\mathcal{X}}$, such that $f$ and $g$ share a common latent space $\mathcal{Z}$ by virtue of having a common encoder $e(\cdot)$ (Fig. 2).

Since $\mathcal{S}_{i,in}$ is a subset of the range of $e$, we have $\mathcal{S}_{i,in} \subseteq \mathcal{Z}$. This further implies that we do not have to apply IBP through $g$ and the whole $f$. Instead we only have to apply it through $f_d$, which does not involve the expansion to the pixel space $\mathcal{X}$. We show in the experiment (Section 3) that this is important to have a practically useful $\mathcal{S}_{i,in}$. Therefore, AuditAI alleviates the need of having a perfect generative model. Nevertheless, we still need to learn a proper latent space $\mathcal{Z}$ and thus encoder for verification. In practice, it is still non-trivial to learn a perfect encoder (e.g., classifying face angle). However, we believe this is still much easier than learning a perfect generator (e.g., generating face images of a particular angle). Next, we discuss how we learn latent space and thus encoder that aims to satisfy our goal of auditing AI models.

**Learning the latent space.** There are three requirements for the latent space. First, it should be *semantically-aligned* for us to specify unit tests in this space. A good example is *disentangled* representation (Higgins et al., 2018; Tran et al., 2017), where each dimension of the latent code would correspond to a semantically aligned variation of the original image. As shown in Figure 2, a latent dimension could correspond to the pose variation, and we can select $e_i$ to be the value of this dimension, and $\mathcal{S}_{i,in}$ as the

proper range that the model would be verified for. Second, our model should be *verifiable*, where the latent space is learned such that the corresponding $\mathcal{S}_{i,in}$ is as large as possible given a specification function $F$. This is a form of *certified training* (Zhang et al., 2019b) to improve the verifiability of the model. Finally, the latent space should be able to perform well on the downstream task through $f_d$. Combining these three criteria, our full training criteria combines three losses $L_{\text{gen}}$, $L_{\text{spec}}$, and $L_{\text{task}}$ that would encourage interpretability, verifiability, and task performance respectively. Note that $L_{\text{gen}}$ is not always used in certification of the classifier, when the generative model is trained completely offline with unsupervised data. We c We explain each of the losses below:

**Task Performance ($L_{\text{task}}$).** Let, CE($\cdot$) denote the standard cross-entropy loss. $L_{\text{task}}$ captures the overall accuracy of the downstream task and can be expressed as $L_{\text{task}} = \text{CE}(z_K, y_{\text{true}})$, where $z_K$ denotes the output logits

**Verifiability ($L_{\text{spec}}$).** Let $f_d$ be a feedforward model (can have fully connected / convolutional / residual layers) with $K$ layers $z_{k+1} = h_k(W_k z_k + b_k)$, $k = 0, ..., K-1$, where, $z_0 = e(x)$, $z_K$ denotes the output logits, and $h_k$ is the activation function for the $k^{\text{th}}$ layer. The set of variations $\mathcal{S}_{i,in}$ could be such that the $l_p$ norm for the $i^{\text{th}}$ latent dimension is bounded by $\epsilon$, i.e. $\mathcal{S}_{i,in} = \{z : ||z_i - z_{0,i}||_p \leq \epsilon\}$. We can bound the output of the network $\mathcal{S}_{out} = \{z_K : l_K \leq z_K \leq u_K\}$ through a variant of interval bound propagation (IBP). Let the specification be $F(z, y) = c^T z_K + d \leq 0$, $\forall z \in \mathcal{S}_{i,in}, z_K = f_d(z)$.

For a classification task, this specification implies that the output logits $z_K$ should satisfy a linear relationship for each latent space variation such that the classification outcome $\arg\max_i z_{K,i}$ remains equal to the true outcome $y_{\text{true}}$ corresponding to $z_0$. To verify the specification, IBP based methods search for a counter-example, which in our framework amounts to solving the following optimization problem, and checking whether the optimal value is $\leq 0$.

$$\max_{z \in \mathcal{S}_{in}} F(z, y) = c^T z_K + d \quad s.t. \quad z_{k+1} = h_k(W_k z_k + b_k) \quad k = 0, ..., K-1 \tag{2}$$

By following the IBP equations for bound-propagation, we can obtain upper and lower bounds $[\underline{z}_K, \overline{z}_K]$ which can be used to compute the worst case logit difference $\overline{z}_{K,y} - \underline{z}_{K,y_{\text{true}}}$ between the true class $y_{\text{true}}$ and any other class $y$. We define $\hat{z}_K = \overline{z}_{K,y}$ (if $y \neq y_{\text{true}}$); $\hat{z}_K = \underline{z}_{K,y_{\text{true}}}$ (otherwise). Then, we can define $L_{\text{spec}}$ as $L_{\text{spec}} = \text{CE}(\hat{z}_K, y_{\text{true}})$. In practice, we do not use IBP, but an improved approach CROWN-IBP (Zhang et al., 2019b) that provides tigher bounds and more stable training, based on the AutoLiRPA library (Xu et al., 2020a). We mention details about the specifics of this in the Appendix.

**Interpretability ($L_{\text{gen}}$).** Finally, we have a loss term for the generative model, which could be a VAE, a GAN, or variants of these. If the model is a GAN, then we would have an optimization objective for GAN inversion. Let's call the overall loss of this model $L_{\text{gen}}$. Note that $L_{\text{gen}}$ could also encapsulate other loss objectives for more semantically-aligned disentanglement. Having a semantically-aligned latent space is important to not only the interpretability but also the accuracy of our verification results. We discuss further limitations of our approach in Section 6. Typically we would train this generative model on a large amount of unlabelled data that is not available for training the classifier we are verifying. We discuss specifics about this for each experiment in the Appendix.

## 2.4 Deployment

Based on the perturbation radius $\epsilon$ and the nominal $z_0$, we have different ranges $[z_0^n - \mathbb{1}_i\epsilon, z_0^n + \mathbb{1}_i\epsilon]$ for the verified error at the end of training the classifier. Here, $z_0^n = e(x^n)$ $\forall$ $n = 1, ... N$ ($N$ is the number of datapoints in the dataset) in the dataset $\mathcal{D}_{\text{train}}$ and $\mathbb{1}_i = 1$ for the $i^{\text{th}}$ dimension (corresponding to the $i^{\text{th}}$ unit test) and is 0 otherwise. We can convert this per-sample bound to a global bound for the $i^{\text{th}}$ latent code, $[\overline{z}_i, \underline{z}_i]$ by considering an aggregate statistic of all the ranges, for example

$$[\overline{z}_i, \underline{z}_i] = [\mu(\{z_{0,i}^n - \mathbb{1}_i\epsilon\}_{n=1}^N) - \sigma(\{z_{0,i}^n - \mathbb{1}_i\epsilon\}_{n=1}^N), \mu(\{z_{0,i}^n + \mathbb{1}_i\epsilon\}_{n=1}^N) + \sigma(\{z_{0,i}^n + \mathbb{1}_i\epsilon\}_{n=1}^N)]$$

Here, $\mu(\cdot)$ denotes the mean and $\sigma(\cdot)$ denotes the standard deviation. Note that this is a global bound on the range of variations for only the latent dimension corresponding to the unit test, and does not concern the other dimensions. During deployment, the end-user receives the model spec-sheet containing values of $[\overline{z}_i, \underline{z}_i]$ for each unit test $i$ and results with different values of perturbation radius $\epsilon$. Now, if the end-user wants to

deploy the model on an evaluation input $x_{eval}$, they can encode it to the latent $\mathcal{Z}$ space to get code $z_{eval}$ and check whether $z_{eval,i} \in [\bar{z}_{0,i}, \underline{z}_{0,i}]$ in order to decide whether to deploy the model for this evaluation input.

## 3 Experiments

We perform experiments with four datasets in different application domains. We choose the datasets ImageNet, LSUN, FFHQ, and CheXpert where current state of the art generative models have shown strong generation results (Karras et al., 2019; 2017; Brock et al., 2018). We show an illustration of what the generated images under latent perturbations look like in Fig 3. Through experiments, we aim to understand the following:

1. How do the verified error rates vary with the magnitude of latent perturbations?
2. How does AuditAI compare against pixel-perturbation based verification approaches?
3. How does AuditAI perform during deployment under domain shift?

### 3.1 ImageNet

ImageNet contains 1000 different classes of objects. We consider a BigBiGAN model (Donahue & Simonyan, 2019) as the base generative model such that the latent space is class conditional, and we can obtain latent codes corresponding to specific classes. So, latent perturbations correspond to variations in different images within the same class. The specification for verification is that the classifier outputs the correct class for all latent perturbations $\epsilon$. This is an unit test because we can verify with respect to each class separately. Table 1 shows results for different $\epsilon$ on the test set. The results confirm the intuition that increasing $\epsilon$ increases the verified error due to wider latent perturbations. The fraction of test images within the range described in section 2.4, for $\epsilon = 25\%$ in Table 1 is 85.3%.

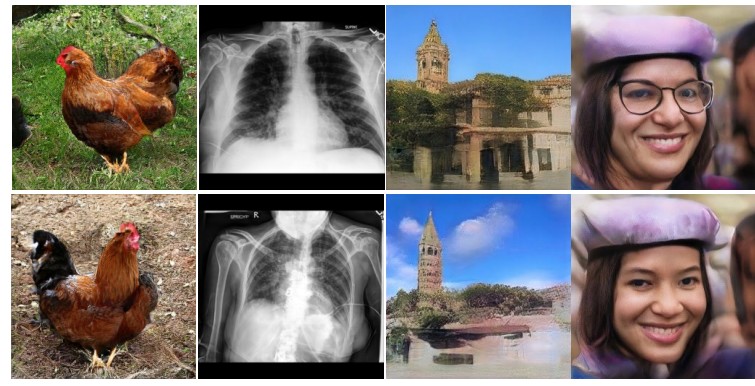

Figure 3: Generated samples from latent space manipulation of features on the ImageNet, CheXpert, LSUN Tower, and the FFHQ datasets. We respectively show two images of a hen looking left and right, chest X-ray images with and without penumonia, Towers with and without vegetation and faces, with and without glasses. These are examples of features that can be controlled through latent space manipulations.

In Table 2, we compare a pixel-based robust training approach with our framework, both trained with the same underlying algorithm CROWN-IBP (Zhang et al., 2019b). For the same verified test error, we have the corresponding latent perturbation for AuditAI (call it $\epsilon_1$) and the pixel perturbation for the pixel-based approach (call it $\epsilon_2$). Based on $\epsilon_2$ we compute the bound for the intermediate latent node, $\epsilon_{21}$. Since comparing $\epsilon_1$ and $\epsilon_{21}$ directly may not be meaningful due to different nominal values of the latent code, we consider the fraction $\epsilon/z_{nom}$ and plot the resulting values in Table 2. We observe that $\epsilon_1/z_{nom_1} >> \epsilon_{21}/z_{nom_{21}}$ indicating tolerance to much larger latent variations. Since a wider latent variation corresponds to a wider set of generated images (for the same decoder), AuditAI effectively verifies over a larger range of image variations, compared to perturbations at the level of pixels. In addition, when we translate the latent perturbations of AuditAI to the pixel space of the generative model (For translation to the pixel-space, we simply use the decoder to map the upper and lower bounds in the latent space to the output space), we obtain on average 20%, and 17% larger variations measured as $L_2$ distances compared to the pixel-based approach for a verified error of 0.88 and 0.50 respectively.

### 3.2 CheXpert

CheXpert (Irvin et al., 2019) contains 200,000 chest radiograph images with 14 labeled observations corresponding to conditions like pneumonia, fracture, edema etc. We consider a PGGAN (Karras et al., 2017) as the base generative model such that the latent space is class-conditional. We consider three separate binary classification tasks with respect to the presence or absence of three conditions: pneumonia, fracture, and

Table 1: Verified Error fraction on a subset (100 classes) of the **ImageNet** dataset. Results are on the held-out test set of the dataset and correspond to training the classifier with the same $\epsilon$ during training. S.D. is over four random seeds of training. **100 classes** is the average value of 100 unit tests over all the 100 classes while **min class** and **max class** respectively correspond to the minimum and maximum verified errors for unit tests over each class. Alongside each $\epsilon$ we represent the % with respect to the nominal value it corresponds to (the nominal value of $\epsilon$ is 0.2, so $\epsilon = 0.05$ is 25% of 0.2). Lower is better.

| **ImageNet** | $\epsilon = \mathbf{0.00(0\%)}$ | $\epsilon = \mathbf{0.05(25\%)}$ | $\mathbf{0.1(50\%)}$ | $\mathbf{0.15(75\%)}$ | $\mathbf{0.2(100\%)}$ | $\mathbf{0.4(200\%)}$ |
|---|---|---|---|---|---|---|
| **100 classes** | $0.15\pm0.06$ | $0.20\pm0.05$ | $0.51\pm0.06$ | $0.57\pm0.05$ | $0.89\pm0.04$ | $0.94\pm0.05$ |
| **min class** | $0.03\pm0.01$ | $0.07\pm0.06$ | $0.33\pm0.02$ | $0.41\pm0.05$ | $0.58\pm0.05$ | $0.86\pm0.07$ |
| **max class** | $0.36\pm0.03$ | $0.41\pm0.04$ | $0.68\pm0.04$ | $0.73\pm0.06$ | $0.93\pm0.05$ | $0.99\pm0.05$ |

Table 2: Comparison of pixel-based variation for CROWN-IBP and `AuditAI` on 100 classes of the **ImageNet** dataset. We tabulate the values below relative to the nominal values of the corresponding latent codes (i.e. $\epsilon/z_{nom}$) and observe that $\epsilon_1/z_{nom_1} >> \epsilon_{21}/z_{nom_{21}}$. As a wider latent perturbation corresponds to a wider set of generated images, `AuditAI` effectively verifies over a larger range of image variations. Higher is better.

| **Verified Error** | **0.88** | | **0.50** | | **0.20** | |
|---|---|---|---|---|---|---|
| **Method** | Pixel $\epsilon_{21}$ | `AuditAI` $\epsilon_1$ | Pixel $\epsilon_{21}$ | `AuditAI` $\epsilon_1$ | Pixel $\epsilon_{21}$ | `AuditAI` $\epsilon_1$ |
| **% Latent variation** | 0.21% | 20% | 0.14% | 12% | 0.08% | 9.1% |

edema. We consider the entire dataset for training the generative model, but for the classifiers we only use positive (presence of the condition) and negative (absence of the condition) data with an 80-20 train/test split. Table 4 shows results for different $\epsilon$ on the test set for each classifier. The fraction of test images within the range described in section 2.4, for $\epsilon = 25\%$ in Table 3 is 89.3%.

As a sanity check, in order to determine whether the generative model behaves correctly with respect to class conditional generation, we consider an off-the-shelf classifier for pneumonia detection trained on real images of pneumonia patients (Rajpurkar et al., 2017). We randomly sample 1000 images from the generative model for each value of $\epsilon = 0.5, 1.0, 1.5, 2.0$ and evaluate the accuracy of the off-the-shelf classifier on these generated images. We obtain respective accuracies $85\%, 84\%, 81\%, 75\%$ indicating highly reliable generated images.

**Deployment under domain-shift.** In order to better understand the process of deploying the trained model on a dataset different from the one used in training, we consider the dataset NIH Chest X-rays (Wang et al., 2017). This dataset contains (image, label) pairs for different anomalies including pneumonia and edema. We perform certified training on CheXpert for different values of $\epsilon$ (results of this are in Table 4) and deploy the trained model to the end-user. The end-user samples 5000 images from the NIH Chest X-rays dataset, encodes them to the latent space with trained encoder $e(\cdot)$ and uses the trained downstream task classifier $f_d(\cdot)$ to predict the anomaly in the image. In Table 3, we list the precision, recall, and overall accuracy among the 5000 images for two conditions, presence of pneumonia and edema. We observe that the results for `AuditAI` are on average 5% higher than the pixel-based approach. This intuitively makes sense because from Table 2 and section 3.1, we see that `AuditAI` covers about 20% larger variations in the pixel space and so the odds of the images at deployment lying in the verified range are higher for `AuditAI`.

### 3.3 FFHQ

Flicker Faces HQ (Karras et al., 2019) contains 70,000 unlabeled images of normal people (and not just celebrities as opposed to the CelebA (Liu et al., 2015) dataset). We train a StyleGAN model (Karras et al., 2019) on this dataset and use existing techniques (Bau et al., 2020a; Shen & Zhou, 2020; Shen et al., 2020) to identify directions of semantic change in the latent space. We identify latent codes (set of dimensions of $\mathcal{Z}$) corresponding to `expression (smile)`, `pose`, and `presence of eyeglasses`. We define the task for the classifier to be detection of whether eyeglasses are present and define unit tests to be variations in the latent code with respect to expression and pose (while keeping the code for eyeglasses invariant). Table 5 shows the results of these unit tests corresponding to different latent perturbations $\epsilon$. We see that the verified error is

Table 3: **Domain shift study**. Evaluation on the **NIH Chest X-Rays** dataset, after being trained on the CheXpert dataset. We perform certified training of the models with $\epsilon = 1.0$ on CheXpert and deploy it on 5000 images of a different dataset to quantify evaluation performance under potential domain shift. We tabulate the values of precision, recall, and overall accuracy. Higher is better.

| Method | Pixel-based | | | AuditAI | | |
|---|---|---|---|---|---|---|
| Metric | Precision | Recall | Accuracy | Precision | Recall | Accuracy |
| Pneumonia | 0.78±0.03 | 0.74±0.04 | 0.80±0.03 | 0.83±0.04 | 0.78±0.05 | 0.84±0.05 |
| Edema | 0.79±0.05 | 0.75±0.03 | 0.81±0.04 | 0.84±0.04 | 0.79±0.03 | 0.86±0.04 |

Table 4: Verified Error fraction on the **CheXpert** dataset. Results are on the held-out test set of the dataset and correspond to training the classifier with the same $\epsilon$ during training. S.D. is over four random seeds of training. Each row is an unit test as it corresponds to verifying a classifier for a particular condition. Alongside each $\epsilon$ we represent the % with respect to the nominal value it corresponds to (the nominal value of $\epsilon$ is 2.0, so $\epsilon = 0.5$ is 25% of 2.0). Lower is better.

| Unit Test | $\epsilon = 0.0(0\%)$ | $\epsilon = 0.5(25\%)$ | $\epsilon = 1.0(50\%)$ | $\epsilon = 1.5(75\%)$ | $\epsilon = 2.0(100\%)$ | $\epsilon = 4.0(200\%)$ |
|---|---|---|---|---|---|---|
| Pneumonia | 0.01±0.01 | 0.02±0.01 | 0.12±0.03 | 0.24±0.05 | 0.43±0.04 | 0.51±0.05 |
| Edema | 0.01±0.01 | 0.03±0.03 | 0.14±0.04 | 0.22±0.04 | 0.41±0.03 | 0.55±0.06 |
| Fracture | 0.02±0.01 | 0.04±0.02 | 0.14±0.03 | 0.22±0.05 | 0.42±0.02 | 0.52±0.03 |

upto 20% for over 50% larger variations compared to nominal values (Table 5 second column, first two rows). The fraction of test images within the range described in section 2.4, for $\epsilon = 25\%$ in Table 5 (FFHQ) is 86.4%. Based on this spec-sheet table, during deployment, the end-user can check whether the encoding of input image lies in the desired accuracy ranges for pose and expression, and then decide whether to use the model.

### 3.4 LSUN Tower

The LSUN Tower dataset (Yu et al., 2015) contains 50,000 images of towers in multiple locations. Similar to the FFHQ setup above, we train a StyleGAN model on this dataset and use existing techniques (Shen & Zhou, 2020; Zhu et al., 2020) to identify directions of semantic change in the latent space. We identify latent codes corresponding to `clouds`, `vegetation`, and `sunlight/brightness`. We define the task for the classifier to be detection of whether vegetation is present and define unit tests to be variations in the latent code with respect to how cloudy the sky is, and how bright the scene is (while keeping the code for vegetation invariant). Table 5 shows the results of these unit tests corresponding to different latent perturbations $\epsilon$. We see that the verified error is around 18% for over 50% larger variations compared to nominal values (Table 5 second column, last two rows) . The fraction of test images within the range described in section 2.4, for $\epsilon = 25\%$ in Table 5 (LSUN) is 90.2%. Based on this spec-sheet table, during deployment, the end-user can check whether the encoding of input image lies in the desired accuracy ranges for clouds in the sky and brightness of the scene, and then decide whether to use the model.

### 3.5 Attacking AuditAI certified images in the pixel-space

In this section, we analyze what fraction of images certified by AuditAI can be attacked in the pixel-space through adversarial corruptions. We create test examples with varying brightness of the scene, corresponding to the setting in Table 5. These are real variations directly by modifying pixels to changing the brightness of the scene . We evaluate AuditAI trained the same way as in Table 5, but evaluated on this test set after certified training. For $\epsilon = 0.5$, 96 out of 1000 images (only 9.6%) certified by AuditAI can be attacked in the pixel-space, while for $\epsilon = 1.5$, 85 out of 1000 images (only 8.5%) certified by AuditAI can be attacked. We use CROWN-IBP (Zhang et al., 2019b) to determine if an image is attackable in the pixel-space. These results show that the certification provided by AuditAI is robust, and although verification is done in the latent space of a generative model, the results hold under real pixel-perturbations corresponding to brightness variations as well.

Table 5: Verified Error fraction on the **FFHQ** and **LSUN** datasets. For FFHQ, the task is to classify whether eyeglasses are present, and the unit tests correspond to variations with respect to pose and expression of the face. For LSUN Tower, the task is to classify whether green vegetation is present on the tower, and the unit tests correspond to variations with respect to clouds in the sky and brightness of the scene. Results are on the held-out test set of the dataset and correspond to training the classifier with the same $\epsilon$ during training. S.D. is over four random seeds of training. Lower is better.

| FFHQ | $\epsilon = 0.0(0\%)$ | $\epsilon = 0.5(25\%)$ | $\epsilon = 1.0(50\%)$ | $\epsilon = 1.5(75\%)$ | $\epsilon = 2.0(100\%)$ | $\epsilon = 3.0(150\%)$ |
|---|---|---|---|---|---|---|
| **Pose test** | 0.01±0.01 | 0.01±0.02 | 0.20±0.05 | 0.37±0.03 | 0.48±0.03 | 0.53±0.06 |
| **Expression test** | 0.01±0.01 | 0.01±0.02 | 0.13±0.04 | 0.29±0.04 | 0.58±0.01 | 0.56±0.03 |
| **LSUN Tower** | $\epsilon = 0.0(0\%)$ | $\epsilon = 0.5(25\%)$ | $\epsilon = 1.0(50\%)$ | $\epsilon = 1.5(75\%)$ | $\epsilon = 2.0(100\%)$ | $\epsilon = 3.0(150\%)$ |
| **Cloud test** | 0.02±0.02 | 0.02±0.03 | 0.14±0.02 | 0.32±0.05 | 0.44±0.06 | 0.51±0.03 |
| **Brightness test** | 0.01±0.01 | 0.01±0.01 | 0.18±0.03 | 0.35±0.05 | 0.49±0.04 | 0.54±0.03 |

## 4    Human Study: How realistic are the generated images?

We employ Amazon Mechanical Turk (Buhrmester et al., 2016) to better understand the generative model trained for the Chest X-ray dataset and determine the limits of latent space variations for which generations are *realistic*. This study is to determine the maximum value of $\epsilon$ for which generations are realistic to the human eye, such that we verify and report in the spec-sheet values of $\epsilon$ only upto the range of realistic generations. We note that this study is to understand the generative model and not the trained classifier. We employ a similar experimental protocol as (Isola et al., 2017; Zhang et al., 2016) and mention details in the Appendix A.4. We generated 2000 images by sampling latent codes within each $\epsilon$ range of variation. Alongside each $\epsilon$ we represent the % with respect to the nominal value it corresponds to (the nominal value of $\epsilon$ is 2.0, so $\epsilon = 0.5$ is 25% of 2.0) in Fig 4. We see that the humans who participated in our study had close to 50% correct guesses for $\epsilon$ upto 1.0, indicating

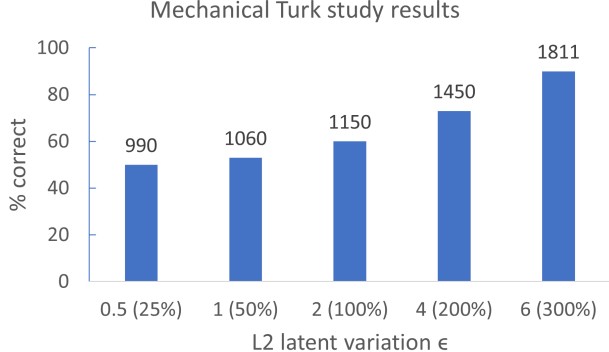

Figure 4: Amazon MTurk study involving 2000 human evaluations for each value of $\epsilon$ in a study of CheXpert Chest X-ray images. We showed pairs of real and generated images to the MTurk workers who chose to take part in our study and asked them to identify which image is real and which is fake (generated). A value close to 50% indicates that the MTurk workers were unable to distinguish the real images from the generated ones, indicating that the generated images looked as good as the real images.

generations indistinguishable from real images. For $\epsilon \geq 4.0$ the correct guess rate is above 70% indicating that the generations in these ranges of latent variations are not realistic, and so the verified accuracy for these ranges should not be included in the spec-sheet.

## 5    Related Work

Multiple prior works have studied a subset of the audit problem in terms of adversarial robustness (Raghunathan et al., 2018a; Dvijotham et al.). Adversarial examples are small perturbations to the input (for images at the level of pixels; for text at the level of words) which mislead the deep learning model. Generating adversarial examples has been widely studied (Xu et al., 2018; Zhang et al., 2019a) and recent approaches have devised defenses against adversarial examples (Xie et al., 2020; Zhang et al., 2019b). Some of these approaches like (Gowal et al., 2018) and (Zhang et al., 2019b) provide provable verification bounds on an output specification based on the magnitude of perturbations in the input. These approaches have largely showed robustness and verification to norm bounded adversarial perturbations without trying to verify with respect to more semantically aligned properties of the input.

Some approaches like (Zhao et al., 2017) uses the generated images / text as semantic adversarial examples with a specific adversarial network and inverter architecture. While, (Yüksel et al., 2021) uses the generations for data augmentation. Another paper (Mirman et al., 2021) has a similar motivation and makes use of generated images for certification. These papers have goals that are different from a unified auditing framework that we have proposed that seeks to audit a classifier by sharing its latent space with a generative model, and considering latent space perturbations, without going to the generated pixel-space for auditing. Other papers (Mohapatra et al., 2020; Ruoss et al., 2020; Balunović et al., 2019; Wong & Kolter, 2020; Qiu et al., 2020; Xu et al., 2020b; Laidlaw & Feizi, 2019) consider semantic perturbations like rotation, translation, occlusion, brightness change etc. directly in the pixel space, so the range of semantic variations that can be considered are more limited. This is a distinction with AuditAI, where by perturbing latent codes directly (as opposed to pixels), the range of semantic variations captured are much larger.

A related notion to auditing is expalainable AI (Gunning, 2017). We note that auditing is different from this problem because explanations can be incorrect while remaining useful. Whereas in auditing, our aim is not to come up with plausible explanations for *why* the model behaves in a certain way, but to quantify *how* it behaves under semantically-aligned variations of the input.

Some papers (Mandelbaum & Weinshall, 2017; Van Amersfoort et al., 2020) have used model confidence and ucnertainty estimates for rejecting out-of-distribution samples at test time, which is a form of auditing. In (Mitchell et al., 2019) and (Gebru et al., 2018), the authors proposed a checklist of questions that should be answered while releasing DL models and datasets that relate to their motivation, intended use cases, training details, and ethical considerations. These papers formed a part of our motivation and provided guidance in formalizing the audit problem for DL models. We provide a detailed survey of related works in section A.6.

## 6 Discussion, Limitations, and Conclusion

In this paper we developed a framework for auditing of deep learning (DL) models. There are increasingly growing concerns about innate biases in the DL models that are deployed in a wide range of settings and there have been multiple news articles about the necessity for auditing DL models prior to deployment[1][2]. Our framework formalizes this audit problem which we believe is a step towards increasing safety and ethical use of DL models during deployment.

We address specific ethical concerns of the framework, `AuditAI` below:

1. `AuditAI` relies on the use of deep generative models, trained on a large set of unsupervised data and the training objective encourages mimicking the training distribution. As such, any bias introduced in data collection will make the generative model generate samples with a similar bias.

2. In the paper, we perform human studies through Amazon Mechanical Turk, in order to determine the image generation quality of the generative model. We have obtained the necessary permissions from our employer prior to performing this study.

3. All the datasets used in the paper for experimentally demonstrating the efficacy of the framework are publicly available standard machine learning datasets - ImageNet, CheXpert, LSUN, FFHQ. To the best of our knowledge, the datasets do not contain any personally identifiable information.

One of the limitations of `AuditAI` is that its interpretability is limited by that of the built-in generative model. While exciting progress has been made for generative models, especially ones that are disentangled and semantically-aligned, the latent dimensions of such generative models still might not accurately capture the desired semantic interpretation. For example, while a latent dimension could generally corresponds to the face angle of an face image, currently formal guarantees of such properties are still open problems. Since our verification depends on the semantic-alignment of latent dimensions, this also affects our verification results. Thus, we believe it is important to incorporate domain expertise to mitigate potential dataset biases and human error in both training & deployment. Currently, `AuditAI` doesn't directly integrate human domain

---

[1]https://www.technologyreview.com/2021/02/11/1017955/auditors-testing-ai-hiring-algorithms-bias-big-questions-remain/, Feb 11, 2021

[2]https://www.wired.com/story/ai-needs-to-be-audited/, Jul 10, 2019

experts in the auditing pipeline, but indirectly uses domain expertise in the curation of the dataset used for creating the generative model.

Another limitation of the current instantiation of `AuditAI` is that many real-world scenarios with richer data such as traffic signs do not easily fit into the current decomposition of semantic variations. As such the framework would require several additional components for accurately modeling such rich semantic variations. Hence, the framework proposed is not a general solution for auditing models in arbitrary scenes, but we still believe it is an important step in the direction because the datasets examnined have been widely used in the vision community for several tasks, and are diverse - covering ImageNet, human faces, towers, and chest X-rays, with practical utility.

Although we have demonstrated `AuditAI` primarily for auditing computer vision classification models, we hope that this would pave the way for more sophisticated domain-dependent AI-auditing tools & frameworks in language modelling and decision-making applications.

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

## A   Appendix

### A.1   Preliminaries: Interval-Bound Propagation

In this section, we provide a preliminary treatment of Interval-Bound Propagation based robust training and verification. Let $f$ be a feedforward classifier model that outputs one of $N$ classes. The input to the network is an image $x_0$, and it has a certain ground truth class $y_{\text{true}}$. The classifier is trained to minimize a certain misclassification loss. The verification problem is to check whether there is some set $\mathcal{S}_{in}(x_0)$ around perturbations of $x_0$ such that the classification output of the model remains invariant from that of $y_{\text{true}}$.

Let $f$ have $K$ layers $z_{k+1} = h_k(W_k z_k + b_k)$, $k = 0, ..., K - 1$, such that $z_0 \in \mathcal{S}_{in}(x_0)$, $z_K$ denotes the output logits, and $h_k$ is the activation function for the $k^{\text{th}}$ layer. If we consider the set of variations to be $\mathcal{S}_{in}(x_0) = \{x | ||x - x_0||_\infty \le \epsilon\}$, then we want to verify the specification that $\arg\max_i z_{K,i} = y_{\text{true}} \ \ \forall \ z_0 \in \mathcal{S}_{in}(x_0)$. We can write this as a general linear specification $c^T z_K + d \le 0 \ \ \forall \ z_0 \in \mathcal{S}_{in}(x_0)$

To verify the specification, IBP based methods search for a counter-example that violates the specification constraint, by solving the following optimization problem and checking whether the optimal value $\le 0$.

$$\max_{x \in \mathcal{S}_{in}} c^T z_K + d \ \ s.t. \ z_{k+1} = h_k(W_k z_k + b_k) \ \ k = 0, ..., K - 1 \tag{3}$$

Since solving this optimization problem exactly is NP-hard (Zhang et al., 2019b), IBP finds an upper bound on the optimal value and checks whether the upper bound $\le 0$. To get an upper bound on $z_K$, we can propagate bounds for each activate $z_k$ through axis-aligned bounding boxes, $\underline{z}_{k,i}(\epsilon) \le z_{k,i} \le \overline{z}_{k,i}(\epsilon)$, where $i$ indexes dimension. Based on this, the specification is upper-bounded by $\overline{z}_{K,y}(\epsilon) - \underline{z}_{K,y_{\text{true}}}(\epsilon)$.

In addition to the verification problem, IBP can be used to perform certified training such that the specification is likely to be satisfied in the first place at the end of training. For this, the overall training loss would consist of two terms: usual classification loss like cross-entropy that maximizes log likelihood $l(z_K, t_{\text{true}})$, and a loss term that encourages satisfiying the specification $l(\hat{z}_K(\epsilon), t_{\text{true}})$. Here, $\hat{z}_{K,y}(\epsilon) = \overline{z}_{K,y}(\epsilon)$ (if $y \ne y_{\text{true}}$) and $\hat{z}_{K,y}(\epsilon) = \underline{z}_{K,y_{\text{true}}}(\epsilon)$ (if $y = y_{\text{true}}$). So, the overall loss for certified training can be represented as:

$$L = \chi l(z_K, t_{\text{true}}) + (1 - \chi) l(\hat{z}_K(\epsilon), t_{\text{true}})$$

As reported in (Gowal et al., 2018; Zhang et al., 2019b; Xu et al., 2020a), to stabilize training, $\chi$ is varied during training such that for a few initial epochs $\chi = 1$ (normal training) and its value is slowly increased so that robust training kicks in gradually.

### A.2   Theoretical result

We show that although `AuditAI` considers perturbations in the latent space that allows a large range of semantic variations in the pixel-space, it retains the theoretical guarantees of provable verification analogous to IBP-based adversarial robustness (Zhang et al., 2019b; Xu et al., 2020a) that consider pixel-perturbations.

**Theorem 1** (**Verification.** The verifier can verify whether the trained model from the designer satisfies the specifications, by generating a proof of the same)**.** *Let $x \in \mathcal{X}$ be the input, $z_0 = e(x)$ be the latent code, $f_d$ be a feed-forward neural network with bounded derivatives, and $\mathcal{S}_{i,in} = \{z : ||z_i - z_{0,i}||_p \le \epsilon\}$ be the set of latent perturbations. Then, the verifier is guaranteed to be able to generate a proof for verifying whether the linear specification on output logits $c^T z_K + d \le 0 \ \ \forall z \in \mathcal{S}_{i,in}, z_K = f_d(z)$ (corresponding to unit test i) is satisfied.*

*Proof.* Revisiting the notation in section 2, consider a network mapping $f : \mathcal{X} \to \mathcal{Z} \to \mathcal{Y}$ (denote by $f_d : \mathcal{Z} \to \mathcal{Y}$ the downstream task) and a generative model mapping $g : \mathcal{X} \to \mathcal{Z} \to \hat{\mathcal{X}}$, such that $f$ and $g$

share a common latent space $\mathcal{Z}$ by virtue of having a common encoder $e(\cdot)$ (Figure 2). Here we provide a proof based on fully-connected feed-forward network architecture for $f_d$, an arbitrary architecture for $f$ and using the Interval-Bound Propagation (IBP) method for obtaining output bounds. For a general network architecture for $f_d$, we refer the reader to a recent library AutoLirpa (Xu et al., 2020a).

Given input $x \in \mathcal{X}$, let $z_0 = e(x)$ be the latent code, and $\mathcal{S}_{i,in} = \{z : ||z_i - z_{0,i}||_p \le \epsilon\}$ be the set of latent perturbations. Let, $z$ be $d-$dimensional. Here, $z_i$ denotes the dimensions(s) of $z$ that are perturbed, such that $z = z_{d-i} \oplus z_i$. Without loss of generality, we can assume all the dimensions in $z_i$ to be contiguous. The verifier can generate a proof by searching for a worst-case violation

$$\max_{z \in \mathcal{S}_{i,in}} c^T z^K + d \quad s.t. \quad z^{k+1} = h_k(W_k z^k + b_k) \quad k = 0, ..., K-1$$

For the sake of simplicity, first consider the case where $j = d$ i.e. all dimensions of $z$ are perturbed. So, $\mathcal{S}_{in} = \{z^0 : ||z^0 - z_0^0||_p \le \epsilon$. Here, we have denoted by $z^0$ the latent code in $\mathcal{Z}$ space, and will use $z^{k+1} = h_k(W_k z^k + b_k) \quad k = 0, ..., K-1$ to denote the subsequent intermediate variables in $f_1 : \mathcal{Z} \to \mathcal{Y}$. Based on Holder's inequality we can obtain linear bounds for the first variable $z^1$, $[\underline{z}^1, \overline{z}^1]$ as follows:

$$\underline{z}^1 = -\epsilon||\underline{W}^0||_q + \underline{W}^0 z^0 + \underline{b}^0; \quad \overline{z}^1 = -\epsilon||\overline{W}^0||_q + \overline{W}^0 z^0 + \overline{b}^0; \quad 1/p + 1/q = 1$$

Here, $|| \cdot ||_q$ denotes computation of the $l_q-$norm for each row in the matrix and the values of $\underline{W}^0, \overline{W}^0$ can be computed with the functions described in Appendix A.1 of (Xu et al., 2020a) depending on the type of non-linearity in $h^0(\cdot)$.

Now, when $j \ne d$ i.e. only $j$ dimensions of $d$ are perturbed, without loss of generality, we can consider all of the $j$ dimensions to be contiguous such that $z^0 = z_{d-j}^0 \oplus z_j^0$. We can then compute $[\underline{z}_j^1, \overline{z}_j^1]$ as follows:

$$\underline{z}_j^1 = -\epsilon||\underline{W}_j^0||_q + \underline{W}_j^0 z^0 + \underline{b}_j^0; \quad \overline{z}_j^1 = -\epsilon||\overline{W}_j^0||_q + \overline{W}_j^0 z^0 + \overline{b}_j^0; \quad 1/p + 1/q = 1$$

We can compute $z_{d-j}^1$ as follows:

$$z_{d-j}^1 = h^0(W^0 z_{d-j}^0 + b^0)$$

Finally, we can obtain $[\underline{z}^1, \overline{z}^1]$ as follows:

$$\underline{z}^1 = z_{d-j}^1 \oplus \underline{z}_j^1; \quad \overline{z}^1 = z_{d-j}^1 \oplus \overline{z}_j^1$$

Now, given the bound $[\underline{z}^1, \overline{z}^1]$ and the relation $z^{k+1} = h_k(W_k z^k + b_k) \quad k = 0, ..., K-1$, we can obtain the bound for $z^K$, $[\underline{z}^K, \overline{z}^K]$ by recursively computing successive bounds as follows:

$$\underline{z}^{k+1} = h_k \left( W \left( \frac{\overline{z}^k + \underline{z}^k}{2} \right) + b - |W| \left( \frac{\overline{z}^k - \underline{z}^k}{2} \right) \right)$$

$$\overline{z}^{k+1} = h_k \left( W \left( \frac{\overline{z}^k - \underline{z}^k}{2} \right) + b + |W| \left( \frac{\overline{z}^k - \underline{z}^k}{2} \right) \right)$$

Here we have assumed that $h_k$ is an element-wise monotonic function, which is is true for for most common actions like ReLU, tanh, sigmoid etc. Using $[\underline{z}^K, \overline{z}^K]$, the verifier can construct an upper and lower bound on the solution of $\max_{z \in \mathcal{S}_{i,in}} c^T z^K + d$

$$\max_{\underline{z}^K < z < \overline{z}^K} c^T z^K + d$$

It is important to note that the proof the verifier generates to check whether the specification holds relies on computing an upper bound $\overline{z}_K$ and checking whether the upper bound $\overline{z}_K < 0$. If the upper bound is not tight, then there may be cases when the true solution $< 0$ (meaning the specification in Theorem ?? is

indeed satisfied) while the upper bound $\overline{z}_K > 0$ (implying the verifier will state that the specification is not satisfied). This form of *err on the side of caution* is a natural consequence of proof by generating a worst case example, and follows directly from IBP.

$\square$

### A.3 Implementation Details

We use a 32-GB V100 GPU for all our experiments. The implementation is in Python with a combination of PyTorch and Tensorflow deep learning libraries. Details about the specific experiments are mentioned below:

**ImageNet.** We use a pretrained BigBiGAN `https://colab.research.google.com/github/tensorflow/hub/blob/master/examples/colab/bigbigan_with_tf_hub.ipynb` as the base generative model, such that the encoder encodes images to the latent space. The downstream classifier from the latent space consists of 5 fully connected layers with ReLU non-linearities. We use ADAM optimizer with a learning rate of 0.0005. For each value of $\epsilon$ $L_2$ perturbation in Table 1, we train for 100 epochs, using the AutoLiRPA library's implementation of CrownIBP `https://github.com/KaidiXu/auto_LiRPA`. The first 5 epochs are standard training for warming up, and the rest of the epochs are robust training. The training time for 100 epochs is about 8 hours.

**CheXpert.** We train a PGGAN `https://github.com/tkarras/progressive_growing_of_gans` on the CheXpert dataset. In order to learn latent embeddings from raw images, we need an inference mechanism. For this, we perform GAN inversion of this pretrained generator, using `https://github.com/davidbau/ganseeing/blob/release/run_invert.sh` to obtain an encoder for projecting images to the latent space. The downstream classifier from the latent space consists of 5 fully connected layers with ReLU non-linearities. We use ADAM optimizer with a learning rate of 0.0005. For each value of $\epsilon$ $L_2$ perturbation in Table 1, we train for 100 epochs, using the AutoLiRPA library's implementation of CrownIBP `https://github.com/KaidiXu/auto_LiRPA`. The first 5 epochs are standard training for warming up, and the rest of the epochs are robust training. The training time for 100 epochs is about 7 hours.

**LSUN and FFHQ.** We train StyleGAN models `https://github.com/NVlabs/stylegan2-ada-pytorch` on the FFHQ and LSUN datasets, and perform GAN inversion using `https://github.com/genforce/idinvert_pytorch` to obtain respective encoders for a disentangled latent space. The downstream classifier from the latent space consists of 5 fully connected layers with ReLU non-linearities. We use ADAM optimizer with a learning rate of 0.0005. For each value of $\epsilon$ $L_2$ perturbation in Table 1, we train for 100 epochs, using the AutoLiRPA library's implementation of CrownIBP `https://github.com/KaidiXu/auto_LiRPA`. The first 5 epochs are standard training for warming up, and the rest of the epochs are robust training. The training time for 100 epochs is about 7 hours each.

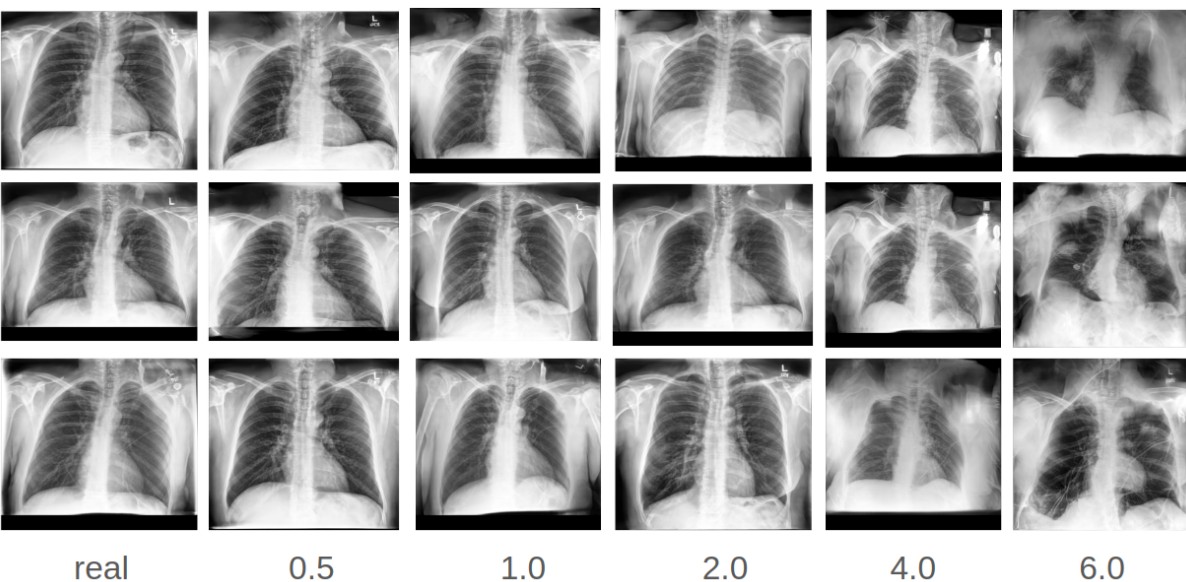

Figure 5: Each column shows images from a aprticular category that are presented in the HITs for the Amazon MTurk Experiment. For each HIT, we show pairs of images such that one image is from the *real* category and the other image is a genertaed image from one of the other categories.

Table 6: In this experiment, we pre-train the classifier through certified training with $\epsilon = 1.0$ and perform verification for different values of $\epsilon$. Verified Error fraction on the **FFHQ** and **LSUN** datasets. For FFHQ, the task is to classify whether eyeglasses are present, and the unit tests correspond to variations with respect to pose and expression of the face. For LSUN Tower, the task is to classify whether green vegetation is present on the tower, and the unit tests correspond to variations with respect to clouds in the sky and brightness of the scene. Results are on the held-out test set of the dataset and correspond to training the classifier with the same $\epsilon$ during training. S.D. is over four random seeds of training. Lower is better.

| FFHQ | $\epsilon = \mathbf{0.5(25\%)}$ | $\epsilon = \mathbf{1.0(50\%)}$ | $\epsilon = \mathbf{1.5(75\%)}$ | $\epsilon = \mathbf{2.0(100\%)}$ | $\epsilon = \mathbf{3.0(150\%)}$ |
|---|---|---|---|---|---|
| **Pose test** | 0.005±0.01 | 0.20±0.05 | 0.39±0.02 | 0.51±0.04 | 0.56±0.05 |
| **Expression test** | 0.005±0.005 | 0.13±0.04 | 0.33±0.04 | 0.57±0.01 | 0.58±0.03 |
| **LSUN Tower** | $\epsilon = \mathbf{0.5(25\%)}$ | $\epsilon = \mathbf{1.0(50\%)}$ | $\epsilon = \mathbf{1.5(75\%)}$ | $\epsilon = \mathbf{2.0(100\%)}$ | $\epsilon = \mathbf{3.0(150\%)}$ |
| **Cloud test** | 0.01±0.01 | 0.14±0.02 | 0.36±0.04 | 0.47±0.06 | 0.52±0.04 |
| **Brightness test** | 0.005±0.01 | 0.18±0.03 | 0.38±0.05 | 0.52±0.06 | 0.57±0.04 |

## A.4 Details about the Amazon MTurk study

Here we provide more details about the setup for the Amazon MTurk study whose results we presented in Fig. 4. We designed each Human Intelligence Task (HIT) to be comprised of 10 pairs of images sequentially presented, such that each pair consists of a real and a generated image. For different values of $\epsilon$ we show examples of images from each of the categories presented in the HITs through Fig. 5.

We recruited participants only from the "Master Worker" category. As defined in `https://www.mturk.com/worker/help`, *"a Master Worker is a top Worker of the MTurk marketplace that has been granted the Mechanical Turk Masters Qualification."* These Workers have consistently demonstrated a high degree of success in performing a wide range of HITs across a large number of Requesters. The participants were paid 0.5 dollars for each HIT (i.e. for 10 pairs of images). Each pair was shown to the participants for 2 seconds, after which the images disappeared from the screen, but they could take as much time as they wanted to decide and form a response. We restricted the number of HITs per participant to 20.

The participants were shown the exact text below: *Choose the more realistic image between options A and B. For each pair, you have 5 seconds to view the image and unlimited time to make the decision. Each task consists of 10 image pairs.*

## A.5 Results of verification with pre-trained classifier

In this experiment, we pre-train the classifier through certified training with $\epsilon = 1.0$ and perform verification for different values of $\epsilon$.

## A.6 Extended Related Works

**Adversarial robustness.** Multiple prior works have studied a subset of the audit problem in terms of adversarial robustness (Carlini et al., 2019; Jia et al., 2019; Gowal et al., 2018; Zhang et al., 2018; Singh et al., 2019; Weng et al., 2018; Singh et al., 2018; Wang et al., 2018; Raghunathan et al., 2018a;b; Dvijotham et al.). Adversarial examples are small perturbations to the input (for images at the level of pixels; for text at the level of words) which mislead the deep learning model. Generating adversarial examples has been widely studied (Zhao et al., 2017; Athalye et al., 2018; Carlini & Wagner, 2017a;b; Goodfellow et al., 2015; Madry et al., 2018; Papernot et al., 2016; Xiao et al., 2019b; 2018b;c; Eykholt et al., 2018; Chen et al., 2018; Xu et al., 2018; Zhang et al., 2019a) and recent approaches have devised defenses against adversarial examples (Guo et al., 2018; Song et al., 2017; Buckman et al., 2018; Ma et al., 2018; Samangouei et al., 2018; Xiao et al., 2018a; 2019a; Xie et al., 2020; Cohen et al., 2019; Gowal et al., 2018; Zhang et al., 2019b). Some of these approaches like (Gowal et al., 2018) and (Zhang et al., 2019b) provide provable verification bounds on an output specification based on the magnitude of perturbations in the input. These approaches have largely showed robustness and verification to norm bounded adversarial perturbations without trying to

verify with respect to more semantically aligned properties of the input. Some papers (Mohapatra et al., 2020; Ruoss et al., 2020; Balunović et al., 2019) consider semantic perturbations like rotation, translation, occlusion, brightness change etc. directly in the pixel space, so the range of semantic variations that can be considered are more limited. This is a distinction with `AuditAI`, where by perturbing latent codes directly (as opposed to pixels), the range of semantic variations captured are much larger. In (Wong & Kolter, 2020), the authors model perturbation sets for images with perturbations (like adversarial corruptions) explicitly such that a conditional VAE can be trained. While, for `AuditAI`, we identify directions of latent variations that are present in pre-trained GANs, and as such do not require explicit supervision for training the generative model, and can scale to datasets like ImageNet and CheXpert, and high dimensional images like that in FFHQ.

**Neural network interpretability.** With the rapid rise of highly reliable generative models, many recent works have sought to analyze the latent space of such models and how the latent space can be manipulated to obtain semantic variations in the output image. (Bau et al., 2021) use text embeddings from CLIP (Radford et al., 2021) to perform directed semantic changes in the generated images, for example introduction of a particular artifact in the room or change in the color of a bird. (Bau et al., 2020a; Shen & Zhou, 2020; Shen et al., 2020) identify latent code dimensions corresponding to pose of the body/face, expression of the face, orientation of beds/cars in the latent space of generative models like PGGAN (Karras et al., 2017) and StyleGAN (Karras et al., 2019). By manipulating these latent dimensions, the output images can be controllably varied. In addition, there have been other papers that sought to understand what the neural network weights are learning in the first place (Bau et al., 2020b; Yeh et al., 2020) in an attempt to explain the predictions of the networks.

**Auditing.** While we proposed a framework for auditing of DL models, a complimentary problem is to audit datasets that are used for training DL models. In (Gebru et al., 2018), the authors proposed a checklist of questions to be answered while releasing datasets that relate to the motivation, composition, collection process, processing, and distribution of the dataset. In (Mitchell et al., 2019), the authors proposed a checklist of questions that should be answered while releasing DL models that relate to their motivation, intended use cases, training details, and ethical considerations. These papers formed a part of our motivation and provided guidance in formalizing the audit problem for DL models.

