# OpenReview forum: "Auditing AI Models for Verified Deployment under Semantic Specifications"
_TMLR — Rejected by TMLR_

### Review · Reviewer_mFzJ · 2022-05-21

**Summary Of Contributions:**

The objective of this paper is to generate specifications that can be checked at inference time to see if an input is within a verified range of operation for a model. This takes the form of an encoder and intervals over the latent space dimensions, and associated verified bounds on model predictions. These are called semantically-aligned unit tests, because the specifications are over the latent space instead of the input (pixels).

The authors derive verified bounds on model predictions with respect to variations in the latent space using IBP. The latent space is shared between a generative model and a classifier through a common encoder. The hope is that this gives semantic-alignment to the latent space and in certain cases, the latent dimensions corresponding to well-defined perturbations in the input space (such as pose and angle of a face or presence of clouds in an image) can be identified to produce more interpretable specifications.

The proposed work is evaluated on four image datasets. It shows that the proposed method is able to achieve larger perturbation bounds for the same verified accuracy than using IBP to train with respect to input space perturbations. For two of the four datasets (FFHQ and LSUN Tower), using off-the-shelf methods they identify latent space dimensions that correspond to interpretable changes like changing the facial expression or changing brightness. Additional experiments are performed to evaluate quality of generated images for chest X-rays, generalization to an independently derived dataset as domain shift, and robustness of a certifiably trained model to an adversarial attack.

**Requested Changes:**

* The authors should cross-check their experimental results and make corrections to the inconsistencies discussed above.
* The authors should carefully edit the paper to make it easier and clearer to the reader. They should fix the incomplete explanations outlined above.

**Strengths And Weaknesses:**

Strengths
----
* The problem of checking whether a model could provide a correct prediction for a given input is practically important and the paper provides an interesting solution to that.
* They also argue how the specifications can be independently verified by essentially using IBP but for verification rather than verified training (as used by the designer of the model).

Weaknesses
----
The paper is not written carefully. It is incomplete and inconsistent in many places:
* The authors claim that "For ImageNet, we show that AuditAI can train verifiably robust models which can tolerate 25%larger pixel-space variations compared to pixel-based certified-training counterparts for the same overall verified accuracy of 88%." but Section 4.1 only reports this to be 20%.
* Table 2 and the discussion in Section 4.1 report "Verified Error" of 0.88 but the statement reproduced above from the introduction claims it to be "verified accuracy".
* Table 1 captions says that $\epsilon = 0.005$ is 25% of 2.0.
* Section 4.3 states that " We see that the verified error is upto 20% for over 50% larger variations compared to nominal values." This is a vague statement: are you saying that the verified error is upto 20% as an absolute value or relative (lower or upper)? I could not place these numbers in Table 5.
* A similar issue to above in Section 4.4. for "We see that the verified error is upto 18% for over 50% larger variations compared to nominal values."
* In Section 2.3, $CE(z_K, y_{true})$ uses $z_K$ without defining it first.
* What is $N$ is Section 2.4?
* What is $z_{nom}$ in Section 4.1?

---

> ### Author Response · Authors · 2022-06-10
> **Author response**
>
> Dear reviewer,
>
> Thank you very much for the helpful and detailed comments on our paper, and for pointing out the typos. We have now corrected them in the revised paper. Here are the modifications based on the reviewer’s comments.
>
> In the introduction, the term should be indeed be verified error and it corresponds to Table 2 in the paper (not Table 1 in section 4.1)
>
> We have corrected the Table 1 caption to make the numbers in the caption consistent with the %
>
> We have clarified the line in section 4.4 to read “We see that the verified error is around 18\% for over 50\% larger variations compared to nominal values (Table 5 second column, last two rows) .” This number is just saying that the average in the second column of Table 5 across the last two rows is upper bounded by 0.18
>
> We have clarified the line in section 4.3 to read “ We see that the verified error is upto 20\% for over 50\% larger variations compared to nominal values (Table 5 second column, first two rows).” This number is just saying that the average in the second column of Table 5 across the last two rows is upper bounded by 0.20
>
> We have now defined $z_K$ next to the definition of $CE(z_K,y_{\text{true}})$
>
> $N$ is the number of datapoints in the dataset which we have clarified now.
>
> By nominal values $z_{\text{nom}}$, we mean the average of the latent code value over all datapoints. We have now clarified this in the paper.

---

> > ### Comment · Reviewer_mFzJ · 2022-06-16
> > **Follow-up comments**
> >
> > Thank you for attending to my comments.
> >
> > I still see the following discrepancies:
> >
> > * Referring to my first comment:
> >
> > _The authors claim that "For ImageNet, we show that AuditAI can train verifiably robust models which can tolerate 25%larger pixel-space variations compared to pixel-based certified-training counterparts for the same overall verified accuracy of 88%." but Section 4.1 only reports this to be 20%._
> >
> > My point was that in the intro, you are saying that your models can tolerate 25% larger pixel-space perturbation. But the following statement in Section 4.1 reports different numbers:
> >
> > "In addition, when we translate the latent perturbations of AuditAI to the pixel space of the generative model, we obtain on average 20%, and 17% larger variations measured as L2 distances compared to the pixel-based approach for a verified error of 0.88 and 0.50 respectively."
> >
> > So are you reporting 25% larger pixel-space variations or 20% larger pixel-space variations?
> >
> > * Referring to another of my comments:
> >
> > _Table 1 captions says that  is 25% of 2.0._
> >
> > I now see that the numbers in column headings have changed by a factor of 10. However, the caption still incorrectly states that $\epsilon = 0.05$ is 25% of _2.0_.  Shouldn't you sayd 25% of _0.2_?
> >
> > Now, coming to the correction of verified accuracy of 88% to verified error of 88% (BTW, I didn't refer to Table 1, I did refer to Table 2 only). I am confused about the implication of this result. You can tolerate 20% (or 25% depending on whether the number in Section 4.1 or the intro is correct) more perturbations for verified error of 88%. But isn't 88% verified error a lot? I believe the perturbation bounds at lower values of verified errors (and analogously higher values of verified accuracy) are more important because they tell us how much perturbations the model can tolerate while being robust enough. As stated in Section 4.1, if you reduce verified error to 50% then you can tolerate 17% more perturbation in pixel space (as opposed to 20% perturbations for 88% verified error). How do the pixel-space perturbations compare for **verified accuracy** of say 88% or more?

---

> > > ### Author Response · Authors · 2022-06-21
> > > **author response**
> > >
> > >  — Thank you for the clarification to the original point. We have now modified the introduction to be consistent with section 4.1(now section 3.1) In particular, it is 20% larger variations. Thanks again for catching this!
> > >
> > > — We have also modified the caption of Table 1 to say 25% of 0.2 Thank you for catching this!
> > >
> > > — Thank you for the point about verified error - we apologize if we misunderstood any of the reviewer’s comments regarding this. Yes, the range of variations that can be tolerated decreases, as the verified error decreases. We have added a column in  Table 2 for verified error of 20% (i.e. verified accuracy of 80%), where the method has around 9% higher tolerance to variations compared to pixel-space perturbation.
> > >
> > > Kindly let us know if there is anything else we can clarify or modify regarding this. Thank you!

---

### Review · Reviewer_i3jG · 2022-05-21

**Summary Of Contributions:**

The central contribution of this paper is a method to test and audit deep learning models before deployment. This is accomplished using a sequence of semantically-aligned tests where each test aims at checking some predefined requirement such as reaching a particular threshold of accuracy with respect to a controlled factor variation in the input space that corresponds to some semantic change. Examples of such factors of variation could be lighting conditions or face angles. The method exploits the interpretability of latent space provided by some generative models. The approach is evaluated on a diverse set of datasets ranging from chest X-rays, Flicker faces, ImageNet, and LSUN.


**Broader Impact Concerns:**

The paper aims at a very important problem of auditing ML models before deployment. Its success will have a significant broader impact.

**Requested Changes:**

It would be helpful to add an evaluation on why the latent space capture semantic factors of variation and the perturbation in the latent space can be lifted to semantically meaningful properties as argued in the beginning of the paper. For example, can we experimentally validation that a face is recognizable as long as the specs on the face have a thickness below some threshold?

Could authors please elaborate on the new claim in Theorem 1 beyond just the application of IBP for a model with the latent space treated as its input?

In order to test the generality and effectiveness of the approach and its reliance on learning semantically meaningful latent space, the reviewer would suggest attempting to do this on rich inputs such as traffic scene datasets (e.g. NuScenes).  It is very difficult to be able to get much interpretability for such rich inputs (which coincidentally are also the ones needing audit of their models).

**Strengths And Weaknesses:**

Strengths

* The paper combines and ports to the latent space two very approaches interval bound propagation and certified training, to address the challenge of auditing ML models.

* The paper presents a very interesting approach to make models robust in the latent space and demonstrates that such models are also robust w.r.t input perturbation.

Weakness:

* A major concern of the reviewer is that the experimental evaluation is not convincing for the central claims of the paper. The paper aims at "semantic" testing. But the experiment result is just checking 1. whether it is robust in latent representation (which would be unsurprising because of the use of CROWN-IBP over the latent space) 2. comparison with pixel-perturbation and 3. domain shift robustness. None of these fully address the main goal of finding semantic failure modes. (1) need not represent semantic failure modes unless the latent space captures semantic information in a disentangled manner, and it is not clear that this is the case. (3) Domain shift can be thought to capture some specific kind of "semantic" failure mode, but this is far short of the claim in the paper and its title.

* Another (minor) concern of the reviewer is that the method is relying on disentangled latest space learning and interval bound propagation which have been found to have their own inherent weaknesses (applicability to broader dataset / scalability). In the absence of the implementation of the paper being available, it is not clear if the approach described in this paper can be readily applied to new models trained on new datasets. The reviewer is skeptical of being able to learn generative models that can balance all components L_gen, L_spec and L_task. Can authors point to what is the specific generative modeling method being used here (reference to paper or implementation)?  Would latent space learned over, say, traffic scenes capture any semantically meaningful information? Given that the goal of the paper was to build semantic unit tests, one naturally expects the experimental section to present evidence that such unit tests can be created algorithmically for new models/datasets.

* The paper lacks any major theoretical contribution, and hence, the reviewer is heavily influenced by the rigor in its empirical evaluation. Theorem 1 is just the restatement of the verification goal. It is not clear why it is written as a theorem. The proof is just IBP ported to the z space. If I think of the model with z space as its input, what is different in this setting from the general IBP?

---

> ### Author Response · Authors · 2022-06-10
> **Author response**
>
> Dear reviewer,
>
> Thank you very much for the helpful and detailed comments on our paper. Please find our responses to specific comments and questions below.
>
> - Thank you for the point regarding the latent space capturing semantic factors of variations. We would like to kindly reference the papers [A,B] that we incorporated in our framework for learning disentangled latent representations. In general, the AuditAI framework can be applicable with any other approach for learning disentangled representations, as potentially better approaches are developed in the future. Empirically, we have shown qualitative results of change in expression for different ranges of variation (in the website linked to the Appendix https://sites.google.com/view/audit-ai#h.x1cwyot2tble) and for quantitative results we have conducted the MTurk human evaluation study for this.
>
> - Regarding the generative models and latent space learning, we have provided details of the models used in section A.3 of the Appendix - these are BigBiGAN, PGGAN, and StyleGAN models that have been shown to be applicable for large diverse datasets.
> Yes, the claim of Theorem 1 is exactly what the reviewer mentioned - but it is not very trivial and doesn’t follow directly from the original IBP result, hence for completeness we have stated it in the paper. If the reviewer suggests we move it to the Appendix, and not emphasize it in the paper, we would be very happy to make this change. Thank you!
>
> - Evaluating on Traffic datasets like NuScenes would be beyond the scope of our work, as it would require several additional components for capturing richer semantic variations. We would like to kindly emphasize that while the datasets we considered are relatively simple compared to NuScenes, they have been widely used in the vision community for several tasks, and are diverse - covering ImageNet, human faces, towers, and chest X-rays that are of practical utility. We would be very happy to emphasize this in the paper, and tone down any claim that the reviewer would consider appropriate.
>
> [A] Jiapeng Zhu, Yujun Shen, Deli Zhao, and Bolei Zhou. In-domain gan inversion for real image editing. In European Conference on Computer Vision, pp. 592–608. Springer, 2020.
>
> [B] Yujun Shen and Bolei Zhou. Closed-form factorization of latent semantics in gans. arXiv preprint arXiv:2007.06600, 2020.

---

> > ### Comment · Reviewer_i3jG · 2022-06-16
> > **Follow-up**
> >
> > The reviewer thanks the authors for the response. Yes, moving the statement in Theorem to appendix would be better. As the authors mentioned, richer data such as traffic signs are not amenable to current decomposition into semantic variations. The reviewer is not sure auditing AI models over ImageNet matches with the goals/claims of the paper described in the abstract and introduction. This is a good work in progress, but the results currently do not back up the central claims of the paper. There are two possibilities - toning down the claims on auditing AI and describing results as they are and discussions on the limitations, or extending the current approach to make it useful for datasets that more strongly relate to AI-safety.

---

> > > ### Author Response · Authors · 2022-06-20
> > > **author response**
> > >
> > > Thanks for the helpful suggestions on rephrasing the claims to make them more consistent with the scope of the paper. We would be happy to make modifications to specific parts of the paper that the reviewer believes would be appropriate. We have now explicitly added more discussions regarding the limitations of the framework (section 6) - and mentioned how it is a small step towards semantic auditing of models, and a lot remains to develop a general auditing framework for AI safety. We kindly request the reviewer to let us know if we should make any more specific changes to the claims of the paper in the abstract/introduction/elsewhere. Thank you very much!

---

> > > > ### Comment · Reviewer_i3jG · 2022-06-26
> > > > **Thanks for the response**
> > > >
> > > > Thank you. The reviewer appreciates the discussion of the limitations. But the overall positioning of the paper and initial claims in the title and early parts of the paper are too broad, and possibly confusing to the audience. The reviewer would request a major rewrite and a toning down of claims in the title and the abstract to reflect that it does not solve the semantic auditing problem but is a "first step" in trying to do so. It would also be useful to justify how this is a first step in the right direction for semantic auditing, and why the limitations are possibly surmountable. The reviewer is fearful that this will be a major revision of the paper.

---

### Review · Reviewer_C1N5 · 2022-05-21

**Summary Of Contributions:**

The paper proposes a new framework for auditing AI models prior to deployment by leveraging recent advances in certified AI. In the proposed framework, an encoder is first used to learn a low-dimensional disentangled representation of high-dimensional images. Each dimension in the latent code corresponds to a semantic feature of the original image. IBP-based certified training of a smaller classifier is performed such that its classification is robust with respect to variations in the latent codes. The authors argue that variations in the latent space yield more meaningful specifications than the pixel-based variations tackled by most existing works. They also show how the verification results can be used for generating a global specification sheet for the model that can be leveraged for improving trust in the prediction of the model during deployment. Experimental evaluation is performed on more complex datasets than the usual MNIST/CIFAR10 datasets considered by existing certified training frameworks.

**Broader Impact Concerns:**

The authors discuss the ethical implications of their work sufficiently in Section 7.

**Requested Changes:**

The authors should address my concerns listed above: compare against related work (1),  discuss accuracy/robustness tradeoff (2), provide missing experimental details (3), and explain the training procedure in more detail (5).

**Strengths And Weaknesses:**

Strengths:
1. Auditing AI models via verification is inherently more expensive than checking whether the model "works" by sampling the model on a few thousand test inputs.  Therefore, a common criticism of certified AI from researchers outside the community is whether such methods can ever be used for obtaining provable guarantees on models for more complex datasets such as Imagenet. The authors show that it is possible to do this via a novel framework that does not have the same limitation as existing works: they require specifying unit tests in high-dimensional pixel space and propagating them through a large classifier mapping inputs from the pixel to the output space.

2. Another advantage of the proposed framework is that the results from local tests can be combined to obtain a global specification sheet for the model that can be used during deployment. This is not possible with the local specifications in the pixel space.

3.  The authors provide a method for automatically deriving acceptable variations in the latent dimensions that can possibly generate more semantic meaningful specifications for testing AI models than pixel-based ones.

Weaknesses:
1. Certification with respect to specifications derived from generative models has been done before in the work of https://arxiv.org/pdf/2004.14756.pdf. They develop specialized methods for certifying generative specifications instead of relying on IBP-based methods. The authors should compare against this work.

2. There is an inherent accuracy/robustness tradeoff associated with certified training. I could not find any details about the accuracy of the classifier f in both the main paper and the appendix. It is unclear if using only 5 fully-connected layers for downstream prediction for complex datasets as done in the current work does not compromise significant accuracy?

3. While the method is novel, I found the experimental results in the different tables to be unclear.  I list the issues below:

(a) The text in Section 2.3 seems to imply that the unit tests are defined with respect to variations in the individual latent dimension, however, the epsilons in the different tables are defined wrt L2-norm which implies that multiple latent dimensions are perturbed. The authors should provide details about the number of dimensions in the latent space that are perturbed in the different settings as well as the dimensionality of the latent space.

(b) Since the size of the neural network affects the cost of certified training. The authors should provide more details about the architecture of f_d, such as the input dimension, number of neurons per layer, etc.

(c) Tables 1, 4, and 5 use nominal values of epsilons but this term is never defined in the paper. How is this computed for the different tables?

(d) The numbers in Table 4.1 can be misleading as they depend on the verifier used. It is possible that if one used a more precise verifier then a wider range can be verified for the pixel-based approaches. The authors should clarify this.

4. The usability of the proposed framework relies on the quality of the latent space. In particular, trusting the specifications in Sections 4.3 and 4.4 requires taking a leap of faith and believing that the latent space satisfies desirable properties such as smoothness which are not formally established by any existing work.


5. The authors mention that they combine the three training losses for classification, robustness, and interpretability. However, the generative model seems to train on a different dataset than used by the classifier in several places. This implies that L_{gen} is not always used during the training of the classifier. The authors should clarify this and provide pseudocode of the training algorithm.

---

> ### Author Response · Authors · 2022-06-10
> **Author response**
>
> Dear reviewer,
>
> Thank you very much for the helpful and detailed comments on our paper. Please find our responses to specific comments and questions below.
>
> - Thank you for pointing out the very relevant paper [A] This paper does have similar motivation as our paper with a main difference that we want to verify specifications with respect to semantic variations for a classifier, using a generative model as a bridge such that the encoder of the classifier and of the generative model are the same. On the other hand, [A] makes use of generated images for certification. Another difference is the use of a CrownIBP based verification scheme in our paper that makes fewer assumptions on the structure of the input space and neural activations compared to [A]. Because of fewer assumptions, shared encoder with the generative model, and certification directly in the latent space, we are able to scale to much larger and higher resolution datasets compared to [A]. We have now added these comparisons in the related works. Thanks again for pointing out this paper.
>
> -Thank you for the point regarding accuracy/robustness tradeoff. We have included them in the respective tables in the paper ($\epsilon=0$). We had omitted them earlier for clarity, since we were primarily interested in observing performance with respect to perturbations. Kindly note that the standard setting results are comparable to $\epsilon=25%$ indicating that the performance-robustness tradeoff is not significant. Also, note that encoder which does a lot of the heavy lifting for representation learning  is shared with the generative model, so, a smaller network $f_d$ to map from latent codes to outputs is sufficient, without compromising accuracy.
>
> - Clarifications 3a) The total latent dimensions in all the experiments are 512 and 10 latent latent dimensions correspond to variation with respect to a particular attribute, for each row in the respective Tables.
>
> - Clarifications 3b) We have provided details about the architecture of f_d in the Appendix A.3 of the paper. The input dimension is same as the latent code dimension (i..e 512d) and there are 5 fully-connected layers (neurons 256, 128,64 per layer) with ReLU non-linearities.
>
> - Clarifications 3c) By nominal values, we mean the average of the latent code value over all datapoints. We have now clarified this in the paper.
>
> - Clarifications 3d) Apologies, but there might be a typo in the reviewer’s comment, as Table 4.1 doesn’t exist in the paper. We would be happy to clarify this.
>
> - Clarification 5) We have now clarified this in section 2.3 of the paper. For ImageNet results, we use a pretrained BigBiGAN that was trained on the entire ImageNet dataset and the classifier certification is on 100 classes as mentioned in the paper. For CheXpert,LSUN, FFHQ, the generative models are trained on the entire datasets (mentioned in Appendix A.3) while the certified training of classifier is for the categories that we have in the respective Tables of results. We have now clarified this in the paper.
>
> [A] Mirman, M., Hägele, A., Bielik, P., Gehr, T., & Vechev, M. (2021, June). Robustness certification with generative models. In Proceedings of the 42nd ACM SIGPLAN International Conference on Programming Language Design and Implementation (pp. 1141-1154).

---

> > ### Comment · Reviewer_C1N5 · 2022-06-16
> > **Thanks for your answers. A further question**
> >
> > Dear Authors,
> >
> > I read the revised version and it looks clearer than before. However, I find the comparison with pixel space perturbations in Section 4.1 a bit convoluted and some details are missing. Specifically:
> >
> > 1. How are the pixels perturbed in the image space? Is it L2-based perturbation? If yes, then what makes it a suitable candidate? It is possible that the verified error and variations in the latent space for a different perturbation model (e.g., adversarial patches) might yield better baseline results.
> >
> > 2. "Since a wider latent variation corresponds to a wider set of generated images", this sounds like an assertion and I am not sure if this always holds. Doesn't it depend on how the latent space is mapped to the image space by the decoder? Are you using the same decoder in both cases?
> >
> > 3. "When we translate the latent perturbations of AuditAI to the pixel space of the generative model", how is this translation done? Do you use IBP or exact solver or sampling?
> >
> > 4. The numbers in Table 2 are epsilon values or epsilon values divided by nominal values? Further, the introduction mentions, "For ImageNet, we show that AuditAI can train verifiably robust models which can tolerate 25%larger pixel-space variations compared to pixel-based certified-training counterparts for the same overall verified error of 88% (Table 2).", the numbers in the Table do not validate this claim.

---

> > > ### Author Response · Authors · 2022-06-21
> > > **author response**
> > >
> > > Thank you for the additional questions. Please find our response and pointers to edits below
> > >
> > > - For the pixel-perturbation approach, we followed the CROWN-IBP procedure (with L2 norm perturbations), for consistency with AuditAI that uses a similar strategy as CROWN-IBP but for latent variations.
> > >
> > > - Yes, the decoder is the same for both the comparisons, so we believe the statement makes sense in this case. We are happy to modify it in case the reviewer suggests a re-phrasing to convey it more clearly.
> > >
> > > - For translation to the pixel-space, we simply use the decoder to map the upper and lower bounds in the latent space to the output space. We have now explicitly mentioned this in section 4.1 (now section 3.1) of the paper.
> > >
> > > - Yes, the numbers in Table 2 are epsilon values divided by nominal values, as per the description in section 4.1 (now section 3.1) We have also modified the line in the introduction to be consistent with Table 2 (20% larger variations)
> > >
> > > Kindly let us know if there is anything else we can modify or elaborate upon. Thanks a lot!

---

### Review · Reviewer_QQA1 · 2022-05-28

**Summary Of Contributions:**

One major problem of deploying deep learning models in the real world is that there are no specifications of the performance guarantees of the models. To address this problem, this work introduces semantically-aligned unit tests, to test the fulfillment of predefined specifications. These unit tests are achieved through the manipulation of the latent space of generative models. Additionally, this work proposes AuditAI a certified training method, which incorporates the latent space of a generative model in the training process. Specifically, the latent space is perturbed to incorporate semantical perturbations into the training process. The authors successfully show how AudiAI can be used to audit deep learning models and present experimental results on various datasets.

**Broader Impact Concerns:**

This work does not raise any ethical concerns from my judgment. The authors include a sufficient discussion section.

**Requested Changes:**

I suggest the authors consider and address the points outlined in my weakness section:
1. Evaluation of baseline models (standard, robust).
2. Data and resource-hungry generative models.
3. Control study for a differing generative model.
4. Distinction of AuditAI to natural adversarial examples.
5. Clarification regarding the exact definition of AuditAI.
6. Clarification regarding the statement that AuditAI “bridges the gap between. semantically-aligned formal verification and scalability.
7. Additional evaluations on perturbations in the pixel space (corruptions, adversarial examples).

**Strengths And Weaknesses:**

## Strengths
1. Auditing AI models is an important problem, for which currently no gold standard solution exists. This work contributes toward this goal.
2. AuditAI was evaluated on several datasets from various domains with large image sizes.
3. I appreciate the comment in the discussion that “any bias introduced in data collection will make the generative model generate samples with a similar bias.” This is important to highlight.

## Weaknesses
1. Currently an evaluation of baselines is missing since only the results for the AuditAI models are presented. An evaluation of a “standard” model would be interesting. I am referring to a standard model, as one which was simply trained on the respective dataset. Additionally, a nice-to-have baseline would be the evaluation of a robust model. Here two variations can be tested: (a) a model robust to adversarial examples (i.e. adversarially trained model) (b) a model robust to natural occurring corruptions (e.g. a model trained with DeepAugment and/or AugMix).
2. One drawback of this technique is, that a generative model is required. Commonly, these models require large training data and resources. Hence, it can be unfeasible to train or obtain such a model in the real-world, since the amount of data or the computational resources might not be available.
3. The same generative model is used during training and during evaluation, hence it is not surprising that the model is robust against feature space perturbations on this model. As a control study, the authors could perform a performance evaluation with a separately trained generative model. Another variation to this would be, that the generative model used during training is obtained through different images than the one used for model evaluation, while all the images come from a similar distribution.
4. The authors mention several times, that adversarial examples are not semantically aligned, since they are perturbations in the pixel-space. I would like to highlight that also semantically-aligned adversarial examples exist [A, B, C, D, E]. Hence, I would be interested in an evaluation of AuditAI against semantically-aligned adversarial examples. Further, how is AuditAI different from such semantically-aligned adversarial examples? The approaches proposed in [A, B] appear to be very similar to the proposed AuditAI, which limits the novelty of this work.
5. It is somewhat ambiguous what AuditAI exactly refers to:
a) The summary of contributions refers to AuditAI as a framework “for auditing deep learning models”.
b) Page 5 refers to AuditAI as the certified training procedure: ”AuditAI can be seen as doing certified training”.
It would be good if the authors could be more specific regarding the exact definition of AuditAI.
6. The abstract states that AuditAI “bridges the gap between semantically-aligned formal verification and scalability”. It is further stated that the authors “address the gap between scalability and interpretability by proposing to verify specifications for variations directly in a semantically-aligned latent space of a generative model”. In my opinion, a further elaboration on the topic is necessary since from these two sentences it does not become clear how the gap between scalability and interpretability is addressed, compared for example to previous methods.
7. In my opinion, the claim that “the results hold under real pixel-perturbations as well.” is an overclaim, since the authors evaluate only brightness as an attack on the pixel space. A more extensive evaluation could have given a clearer picture regarding the robustness properties of AuditAI. Such pixel-space evaluation could include the evaluation under natural occurring corruptions, such as JPEG-compression, blur, pixelation, etc. (ImageNet-C), and adversarial examples generated through attacks such as FGSM, PGD, or AutoAttack.

[A] Generating natural adversarial examples; ICRL 2018
[B] Semantic Perturbations with Normalizing Flows for Improved Generalization; ICCV 2021
[C] SemanticAdv: Generating Adversarial Examples via Attribute-conditioned Image Editing; ECCV 2020
[D] Towards Feature Space Adversarial Attack; AAAI 2021
[E] Functional Adversarial Attacks; NeurIPS 2019

---

> ### Author Response · Authors · 2022-06-10
> **Author response**
>
> Dear reviewer,
>
> Thank you very much for the helpful and detailed comments on our paper. Please find our responses to specific comments and questions below.
>
> - Regarding “standard” model results.  We have included them in the respective tables in the paper ($\epsilon=0$). We had omitted them earlier for clarity, since we were primarily interested in observing performance with respect to perturbations. However we agree with the reviewer that these numbers for standard setting would serve as useful reference. Kindly note that the standard setting results are comparable to $\epsilon=25%$ indicating that the performance-robustness tradeoff is not significant.
>
> - Regarding computationally expensive generative models.  Yes, we agree that training generative models on large datasets is expensive, but kindly note that this a one-time cost, as the generative model once pre-trained on large unsupervised data, is fast for inference or fine-tuning.
>
> - Regarding control study with separately trained generative model. Thank you for emphasizing this - kindly note that we already have results for a similar setting in section 4.2 (and Table 3) where we trained the model on the CheXpert dataset and evaluated it on a different dataset, NIH Chest X-rays.
>
> - Regarding semantic adversarial perturbation papers. Thank you for pointing to the papers [A-E]. We had already cited [A], and have now cited and discussed the rest [B-E]. In particular, [A] and [B] both make use of the generated images, for different things - [A] uses the generated images / text as adversarial examples with a specific adversarial network and inverter architecture. While, [B] uses the generations for data augmentation. Both these papers have goals that are different from a unified auditing framework that we have proposed that seeks to audit a classifier by sharing its latent space with a generative model, and considering latent space perturbations, without going to the generated pixel-space for auditing.
>
> - Clarification about the framework. In this paper, we have denoted the entire framework as AuditAI, of which certified training is a part. We have now updated the line referred to by the reviewer in page 5 to remove any ambiguity.
> Regarding addressing the gap between scalability and interpretability. Thank you for the suggestion to elaborate on this. We specifically address this gap, by relying on latent-space perturbations that correspond to semantically aligned variations - compared to semantic adversarial perturbations referenced by the reviewer, this is scalable because the latent space is much low dimensional compared to the pixel-space.
>
> - Thank you for the point about real-pixel perturbations. We have now emphasized in that result that the results hold under real-pixel perturbations corresponding to brightness variations.

---

### Decision · Action_Editors · 2022-07-01

**Recommendation:** Reject

**Comment:**

This paper introduces unit tests for auditing machine learning models, which is an important and unsolved issue if we want to deploy machine learning models to the real world. They also proposed a certified training method AuditAI, which perturb the input in the latent space to generate realistic counterfactual examples.

4 expert reviewers reviewed this paper, and after significant discussion over multiple rounds with the authors, 3 decided to reject the paper in its current form. A significant amount of detailed issues were solved during the discussion period, however, reviewers still found two major issues:

1) The claim seems to be too broad, it is advised that the authors further tune down their claims to ones that can be backed up by experiments.

2) The experiments are only solid in the latent space perturbations. Several reviewers find that the pixel-space perturbation experiments to be still unsatisfactory, both in terms of how the baseline results were presented and in terms of whether the setup is fair.

Based on the majority opinion of the expert reviewers, the editor thus recommend rejection of this paper. Despite this, the problem this paper is studying is important and authors are encouraged to strengthen their paper and submit to the next venue.